# Designing an Education Database in a Higher Education Institution for the Data-Driven Management of the Educational Process

**Tatiana A. Kustitskaya** [1] , **Roman V. Esin** [1] , **Alexey A. Kytmanov** [2,1,*] and **Tatiana V. Zykova** [1]

1 School of Space and Information Technology, Siberian Federal University, 660041 Krasnoyarsk, Russia; tkustitskaya@sfu-kras.ru (T.A.K.); resin@sfu-kras.ru (R.V.E.); tzykova@sfu-kras.ru (T.V.Z.)
2 Institute for Advanced Technologies and Industrial Programming, MIREA—Russian Technological University, 119454 Moscow, Russia
* Correspondence: kytmanov@mirea.ru or aakytmanov@sfu-kras.ru

**Abstract:** During the past two decades, higher education institutions have been experiencing challenges in transforming the traditional way of in-class teaching into blended learning formats with the support of e-learning technologies that make possible the collection and storing of considerable amounts of data on students. These data have considerable potential to bring digital technologies in education to a new level of personalized learning and data-driven management of the educational process. However, the way data are collected and stored in a typical university makes it difficult to achieve the mentioned goals, with limited examples of data being used for the purposes of learning analytics . In this work, based on the analysis of existing information systems and databases at Siberian Federal University, we propose principles of design for a university database architecture that allow for the development and implementation of a data-driven management approach. We consider various levels of detail of education data, describe the database organization and structure, and provide examples of learning analytics tools that can benefit from the proposed approach. Furthermore, we discuss various aspects of its implementation and associated questions.

**Keywords:** learning analytics; digital footprint; education database; digital educational history; learning success prediction; curriculum analytics; data-driven management

## 1. Introduction

### 1.1. Digitalization in Higher Education: Challenges and Opportunities

In recent decades, there has been a dramatic acceleration in the pace of the development and implementation of new technologies, although various gaps persist in terms of implementation in different parts of the world [1]. This rapid technological change, together with the COVID-19 pandemic, has resulted in the explosive growth of digital infrastructure, affecting nearly every area of economic and social life. The amounts of data collected, stored, and processed have increased significantly. However, only a limited portion of these data is used for decision making, usually depending on the data culture in a given company [2–4]. Generally speaking, the active involvement of data-driven management technologies in our daily lives is something that we can expect in the near future [5,6].

Global trends in higher education show a continuous increase in the gross student enrollment rate [7]. In general, an increase in the number of students simultaneously enrolled in a certain higher education institution (HEI) yields an increase in the number of academic staff members [8]. This generally applies to universities with well-known brands, many of which have been expanding in terms of the variety of educational programs they offer. At the same time, top universities usually have better access to technology and a better ability to embrace technological change, although it takes time and effort to do so.

At present, HEIs, as well as online learning platforms, use a variety of tools to collect education-related data. Usually, a considerable portion of these data stays within its traditional "area of use". For instance, data on how a student masters a certain academic course are used by the course instructor to monitor the student's performance and assist them in the successful completion of the course. Such data can also be used by a specific kind of service, so-called early warning systems [9], which are aimed at alarming students whose performance drops below a certain level and poses a risk in terms of the successful completion of the course. At the same time, upon completion of the course, the only piece of data that is used "outside the course" at the administrative level is the final course grade, which is sufficient for maintaining student records and issuing academic transcripts upon graduation. However, based on these scarce data, it is very difficult to judge student experience throughout their studies.

At the university level, student-related, education-related, and learning-related data are used mostly for the purpose of information exchange between various university services, maintaining a fluent administration process. Nevertheless, these data have considerable potential and can be used, for instance, to minimize the number of students unsatisfied with their current studies or failing to complete their educational programs. Implementing such practices can bring the management of the educational process to a new level but requires the investment of considerable amounts of resources. We believe that university leaders and executive teams who manage to use available education-related data to respond to the described education challenge can achieve sustainable development in their HEIs and drive them to a better future, staying ahead of their competitors.

### 1.2. An Overview of Existing Learning Analytics Tools and Practices

Currently, the most obvious and common use of education data in HEIs is for the management of the educational and administrative processes within a university. However, various solutions aimed at optimizing the educational process, based on the analysis of student learning data, are becoming increasingly popular [10].

Both HEIs and massive open online course platforms use learning analytics dashboards to provide learners with concise visualized information on the progress and success of their learning, in addition to providing course instructors with reports on the learning process. The most well-known example is one of the first of such systems, the Course Signal system [11], which visualizes the predicted learning outcomes of students as a traffic light. The forecast is based on course grades, time spent by a student on a certain task and past overall performance. Another example is the My Learning Analytics dashboard [12], which is aimed at helping students to plan their learning throughout a course and understand their course performance with respect to that of their classmates.

To increase efficiency, monitoring tools are often accompanied by recommendations and methodological assistance [13,14]. However, recommendation systems are developed not only to support the educational process within a particular course but also to solve more specific tasks. For instance, in reference [15], the authors introduced a recommendation system for choosing internship placements .

Early warning systems, student relationship engagement systems, and learning support systems that include monitoring, recommendation services, and measures to support at-risk students, together with other instruments, are gaining popularity due to the more extensive use of online learning by HEIs, resulting in a decrease in face-to-face interaction and teacher support. Such systems can be implemented at various levels of educational hierarchy. In references [14,16], learning analytics (LA) solutions were executed at the course level, whereas in references [17,18], the authors presented a review of university-wide learning support systems.

The task of predicting student learning success/failure is one of the key objectives for HEI managerial teams at various levels (university/school/department). The initial setup of the task may vary depending on the managerial team. It can be formulated as a prediction of student learning satisfaction or computation of the probability of a student

successfully completing a certain course, the probability of a student dropping out in the current semester, or the probability of a student successfully completing their studies in the current degree program. Upon solving each task, one may face various difficulties, such as problems in accessing and collecting the data relevant for solving the task.

A considerable amount of research has been conducted on the construction of models to predict freshman dropouts [19,20] as they usually occur at the beginning of training. However, there is a lack of research on general patterns of dropouts. In references [21,22], the authors note that a general model for predicting student dropouts throughout their studies can be useful in determining the main causes of dropouts and dissatisfaction with training over time. There are approaches to solve the problem of monitoring student dropout using machine learning methods with predictors such as academic performance or student socio-demographics [23]. Yet, the static nature of the considered features does not reflect the dynamics of student activity.

### 1.3. The Role of HEI Databases in Data-Driven Management

Education data play an essential role in the LA solutions described above. The effectiveness of instruments built on specific data crucially depends on their completeness, consistency, and diversity. The problem of correct and rational warehousing is a challenging task, for which the solution can significantly improve the performance and quality of the university services [24–27].

Taking into account a significant increase in the amounts of data collected, stored, and used in HEIs, the concept of Data Governance has been introduced into higher education [28,29] and was discussed in relation to LA [30]. As was mentioned in a comprehensive review [31],

> it aims at implementing a corporate-wide data agenda, maximizing the value of data assets in an organization and managing data-related risks.

To address this concept, one should start with questioning what kinds of data are being collected and whether they are sufficient for solving strategic tasks, how data are stored, and how convenient it is to work with them. In other words, the way data are collected and organized may have a crucial impact on the development of tools that can assist in managing numerous aspects of the educational process.

The digital infrastructure of a typical HEI consists of various information systems including a human resource management system, an accounting management system, a document management system, an admission management system, and special-purpose information systems operating with data on research projects, utilities, financial aid, student achievements, etc. The data used by the mentioned information systems are usually stored in various HEI databases. These databases can have different structures, some of which are inherited from their predecessors. The data per se can be inaccurate, inconsistent, or incomplete. Duplications of the same pieces of data in different databases may lead to conflicts during the interaction of information systems with each other. These issues require technical assistance and make the process of developing LA solutions more intricate and challenging. On the other hand, the typical attitude of a HEI management teams towards the organization and structure of HEI databases is that it should not be exposed to a wider audience, even for research purposes.

In reference [32], the authors have described an approach to building a comprehensive digital profile of a student that may serve as a keystone for modeling students' learning behavior, designing their digital twin, and building information systems for the data-driven management of the educational process in a HEI. The digital profile of a student consists of two components: a digital personality portrait and a digital educational history. While the digital personality portrait contains socio-demographics, psychographic data, information on cognitive styles, etc., the digital educational history consists of cumulative data on the educational activities and attained learning outcomes of a student.

In this work, we focus on designing a HEI education database (that can be thought of as a set of databases) in such a way that it can collect, store, and operate with HEI data,

avoiding, or reducing to a minimum, the above-mentioned problems. This can significantly simplify the process of implementing the concept of data-driven decision-making in terms of managing an effective educational process in a HEI.

## 2. Materials and Methods

### 2.1. Existing Information Systems and Data Used

An essential part of our studies was based on the analysis of the existing digital infrastructure of Siberian Federal University, the employer of all of the authors of this work.

Siberian Federal University [33] (SibFU), Krasnoyarsk, Russia, one of the ten federal universities in Russia, was founded in 2006 under a governmental program by merging four regional-level HEIs. There are around 25,000 students and around 2200 academic staff members. Ranking 1201–1500 in the world according to the Times Higher Education World University Rankings, it ranks in the first quartile among Russian universities in terms of the main assessed indicators: research, teaching, share of international students, entrepreneurship, etc.

There are two enterprise content management systems (ECM) that are widely spread in Russia and that are being actively used at the university for the administrative and documentary support of the educational process: *Plans* by MMIS Lab [34] and *1C: Document Management* by 1C [35]. *Plans* is by far one of the most common software systems for designing, storing, and working with all kinds of documentation for educational program support. It is used to develop curricula, personal study plans, and course syllabi, as well as to plan workloads. *1C: Document Management* allows one to manage the documentation regarding university staff and student status: accounting, university regulations, academic orders, etc. Both systems are integrated in the SibFU digital infrastructure as *Plans* electronic service [36] and *1C: Document Management* electronic service [37].

Furthermore, SibFU has developed and operates the *My SibFU* corporate social network service [38] for the university students and staff. It was launched in 2013, and since that has undergone several upgrades. It provides students with access to the curricula of the degree programs they master, course syllabi, course textbooks available in the university library, and personal electronic grade books. It allows them to view the details of issued orders regarding the educational process, and to apply for scholarships or competitions based on their achievements. The service also provides students with an opportunity to communicate with teaching staff, manage team work as a part of academic or extracurricular activities, and follow university news and announcements. Since 2010, SibFU has been developing a Moodle-based *e-Courses* learning management system (LMS) [39] to support student learning activity. To date, more than 20,000 courses have been developed and are available on the *e-Courses* platform. We briefly list some other digital services that operate with data directly or indirectly related to students or the educational process: the electronic library service [40], the corporate video conferencing service [41], the electronic admission service [42], and the research achievements tracking service [43].

The School of Space and Information Technology, a subdivision of SibFU, has developed its own information management system [44] operating within the school. It complements the university-wise digital infrastructure and allows school staff to work as a whole: administrative staff (setting up and editing user access rights to certain components of the information system), educational office staff (creating reports, filling in records, issuing orders, notifications, grade sheets, etc.), teaching staff (filling in and signing grade sheets, linking academic groups to electronic courses, viewing notifications), students (viewing grade books, orders, notifications, ordering certificates), and students' legal representatives (viewing attendance, academic performance, academic orders).

### 2.2. Principles of Designing the Education Database

In reference [32], the authors introduced the notion of student digital educational history. It is multidimensional, structured, dynamically updated data on the educational activity and educational results of a student. This can include the data on the student's

performance prior to enrollment (high school diploma grades, contest results, entrance examination scores), final course grades in the past semesters, current performance, and learning behavior data. It can include data on extracurricular activities, e.g., performance on external educational resources. We use this notion as a basis for designing an architecture of the proposed education database.

Following the definition of digital educational history, we suggest the principles which, in our view, may play a key role when developing an education database, namely

- The principle of student centeredness;
- The principle of data continuity;
- The principle of data consistency.

A student is a key stakeholder of education. Indeed, any educational program is designed and any educational activity is held for a student, and most LA tools are directly or indirectly aimed at improving student learning success. Therefore, to reflect the central role of the student in an education database, we suggest following the principle of *student centeredness*, which, in particular, uses a student identifier as a key field connecting student data with all the other tables of the database and storing all student education-related data collected by HEI information systems.

The principle of *data continuity* keeps all dynamically changing student data with appropriate time tags. It ensures that no data are overwritten and become, therefore, inaccessible for the purposes of monitoring and analyzing their educational history. We insist of keeping the intermediate values of student learning-related characteristics as they can be important evidence of student motivation, learning, and learning performance.

For example, by registering not just the final course grades but also the dates and results of all retakes of the exams, we possess information about learning difficulties experienced by a student and the dynamics of their performance. A similar situation arises with educational program as a characteristic of the learning path of a student. If a student transfers from one program to another, it can indicate a change in their interests, difficulties with mastering the previous program, or the higher competitiveness of the subsequent degree program (an analysis of the reasons for the transfer can be conducted on the basis of the other data on a student's learning path). This can provide academic staff with important insights to enhance the educational program design.

The principle of *data consistency* is crucial not only for education-related data but also for data used in various human-related services. Following this principle means maintaining correct data structure and organization which makes it impossible to enter contradicting information, and minimizes duplications and errors due to human factors. This makes using information from the education database easier and more justified for the purposes of LA and data-driven decision-making.

### 2.3. Development of the Database: Key Steps

To conduct this experimental work, the authors were provided with samples of data connected with students, their learning process, and extracurricular activities. Data samples exported in Microsoft Excel Open XML Spreadsheet (XLSX) format were taken from the university electronic services as follows. Admission data and entrance examination results were obtained from the electronic admission service [42]; the general education data (academic group, chosen electives, links to the assigned e-courses) and student grade book data were obtained from the information system [44]; the data on academic orders and change of status of a student were obtained from the *1C: Document Management* electronic service [37]; the data on existing educational programs and their curricula were obtained from the *Plans* electronic service [36]; the data on student activity in e-courses were obtained from SibFU learning management system [39]; and the data on extracurricular activities and achievements in research, arts, and sports were obtained from the corporate social network service [38].

Based on the study of these samples, we got a comprehensive picture of all kinds of data collected by the university for each student including general personal data, the data

on results of admission entrance tests, the data on educational programs implemented at the university, student grades data, the data on the orders regarding any change of status of a student, the data on student activity in the LMS, and the data on extracurricular activities and achievements in research, arts, and sports.

Our next step was the analysis of the structure of data records and identification of the possible issues such as incompleteness, inconsistencies, errors, outdated pieces, or conflicts of data records in different databases.

Based on the principles listed in Section 2.2, and existing experience in the development of LA solutions, we described the organization and structure of the education database in such a way that ensures avoiding most of the listed above drawbacks and makes working with education-related data more convenient.

Next, we described the theoretical framework, architecture, cybersecurity framework, general data protection regulation (GDPR), and ethical framework. We suggested a mockup of the database and listed key steps of its evaluation plan.

## 3. Results

### 3.1. Description of the Levels of Detail of Education Data

Education data produced by various information systems and university services are often of large volume and diverse structure. To effectively use this data in managing the educational process, a competent organization of its storage and administration is required. Taking into account the hierarchy of both educational and administrative structures in a HEI, as well as the complexity of monitoring and managing the educational process, it seems reasonable to organize the storage and access to education data using a hierarchical structure as well. Within the hierarchy of education data, we identify three levels of detail, namely the university level, the education program level, and the course level (Figure 1).

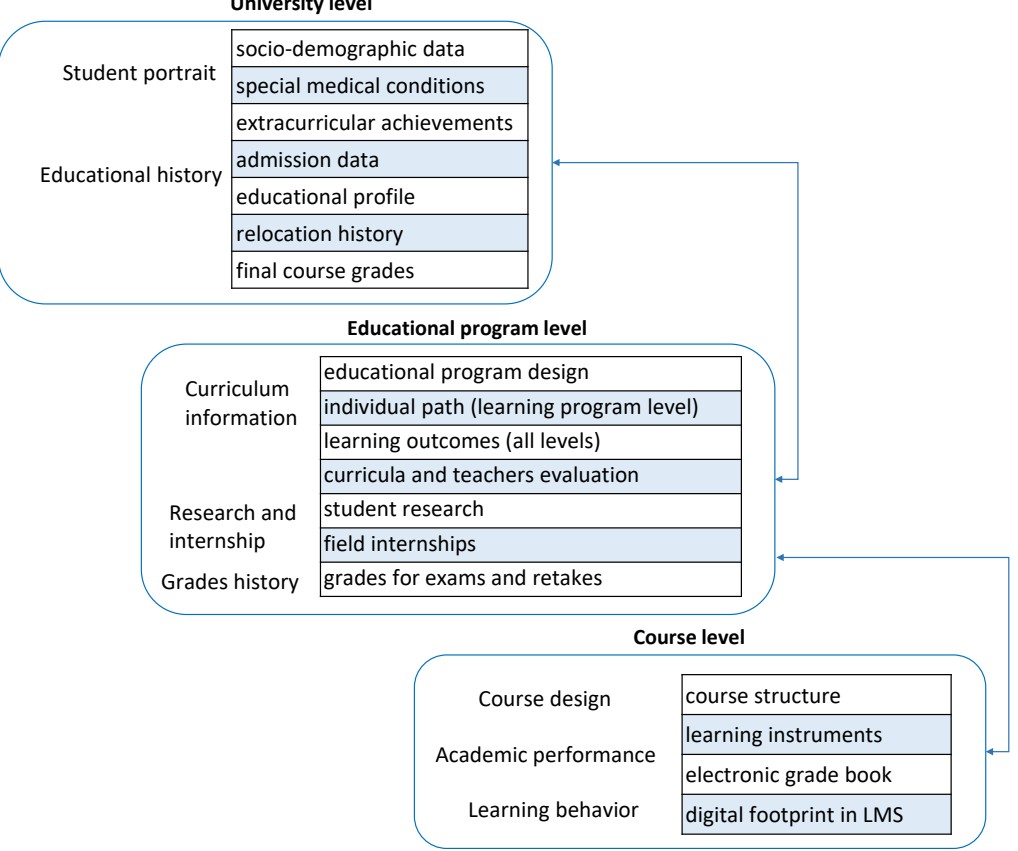

**Figure 1.** Hierarchy of education data.

Each level of education data comes to focus on a certain group of stakeholders. HEI management personnel operate with more general data such as dropout rate, student mobility rate, etc. School/faculty/department administrative staff are more focused on student final course grades and changes in the academic status of a student. Teaching staff usually assess student performance within a certain academic course. Although the pieces of data at different levels of detail are interconnected, representatives of each staff category often work independently of each other with rare occasions of transition to another level not typical to them. This means that existing successful practices of building learning support systems are usually limited to the systems operating withing a certain level of detail. At the same time, problems such as the comprehensive modeling of a student or optimizing the educational process in a HEI require the simultaneous consideration of data at various levels of detail and, therefore, are much more challenging [45].

### 3.1.1. University Level

We start with describing the so-called university level—the most common level of detail used in HEIs. This consists of data necessary for maintaining the educational process and education-related administrative processes for a particular student as well as typical data describing a student. More precisely, the considered data consist of the following categories:

- Socio-demographics;
- Special medical conditions;
- Admission data;
- Final course grades;
- Extracurricular achievements;
- Educational profile;
- Relocation history.

We explore each category more thoroughly. The basic *socio-demographics* consist of data such as a student's name, gender, date of birth (age), place of birth, marital status, citizenship, mother tongue, and level of language proficiency in which the educational program is offered (if different from mother tongue). It can also contain such information as parents' marital status, information about incompleteness of the family, the number of dependent family members, average household income, etc.

*Special medical conditions* are important for implementing inclusive education practices. This category helps to determine whether a student needs classroom accompaniment or extra assistance throughout their studies.

*Admission data* are generally extensively used by university admission services during the admission campaign. These data are passed to the corresponding department registrar's office upon enrollment. Admission data for bachelor studies usually consist of the final grades received in the institution of the previous education level (usually, high school or college) and the results of entrance examinations (or state examinations that count for entrance examinations independently of a certain HEI). Additionally, these data can contain results of higher-level olympiads for high-school students that count towards scores of entrance examinations or, in the case of top-level olympiads, count for them. Usually, the admission data for bachelor studies are rather scarce. Admission data for master-level studies usually include the bachelor studies diploma and the results of entrance examinations, sometimes accompanied with the applicant's list of publications. In certain cases, which usually occur in MBA-type programs for applicants with work experience, there can be portfolio competition instead of entrance examinations. In this case, admission data can include a detailed resume of the applicant, consisting of their education, acquired skills and experience, positions held, etc. It usually goes with a cover letter, motivation letter, and/or an essay connected with the objectives of the educational program for which they applied.

While *final course grades* represent a student's academic performance, extracurricular achievements represent any achievement of a student besides their academic performance:

scientific research results (publications, conference talks, scholarships for researchers), winning awards in olympiads, hackathons, or other types of individual or team competitions (including sports), participation in social events, etc. Usually, extracurricular achievements count towards awarding various kinds of scholarships provided by a HEI.

*Educational profile* reflects the current state of a student at the HEI. It includes their current status: active (undertaking studies at the current moment), on academic leave (including the reasons and the period of leave), inactive (dropped out of their studies) or graduated. Furthermore, it includes a student's current position on their academic track (the list of courses/internships already mastered, timeline for mastering the educational program, name of academic advisor, title of graduation thesis, etc.). Finally, *relocation history data* consist of information on the educational programs/HEIs from which a student has been transferred into their current one.

University services most commonly operate with university-level data. A part of these data, aggregated and depersonalized, is used for statistical reports on a higher level and is utilized by the Ministry of Education, university ranking entities, funding bodies, etc.

3.1.2. Educational Program Level

Educational program level data can be grouped into three blocks.

The *curriculum information block* describes both the curriculum design and the student personal learning path within a certain educational program. It includes information about the following:

- A student's personalized set of educational courses (compulsory courses and the chosen electives, the distribution of workload among the disciplines: number of contact hours, amount of independent work, types of final assessment);
- A list of desired learning outcomes (competences, indicators and descriptors) with each one assigned to a particular course or courses;
- Curriculum evaluation, teaching staff evaluation, and feedback, collected by means of surveys or course assessment questionnaires.

These data provide information about student skills, abilities, and level of satisfaction with their degree program or the teaching staff. Moreover, it is a valuable source of insights about the curriculum coherence and proficiency of the teaching staff. Thus, these data can be used for degree program evaluation as well as for teaching staff assessment.

The *research and internship block* comprises data about

- Student field experience;
- Topics of student research;
- Student publications;
- Participation in conferences;
- Name of scientific advisor;
- Reviews on student research papers and graduation thesis;
- Participation in team projects, including teamwork roles and responsibilities, etc.

These data describe student preferences, proficiency in the research field and levels of soft skills development. It can be used in project team recruitment services or in recommendation systems for internship opportunities or after-graduation recruitment. The names of scientific advisors along with research topics can provide subsequent students with information about opportunities for their own scientific development. These data can be presented in HEI information systems via dashboards.

The *course grade history block* comprises information about the history of the final assessment in a certain course, including:

- Final grades for all courses of the curriculum;
- Dates of exams and retakes;
- Results of all exams and retakes;
- Current number of academic debts.

The educational program level is the main source of data for student success predictive models. Using these data, along with the socio-demographics and history of student dropouts (at the university level), one can predict student performance or the probability of student dropout from the current educational program. The result is preferably presented to the stakeholders responsible for educational program management and student-at-risk supportive measures.

Data at this level are the most relevant source of information for program assessment. For direct evidence of learning, one can use data from the course grade history block to evaluate student performance, and from the research and internship block to evaluate student portfolios, research, and field experience. The results of student and alumni surveys, exit interviews, etc., (the curriculum information block) serve as indirect evidence of educational program quality and can be also taken into consideration.

### 3.1.3. Course Level

Course level is the lowest level of the proposed hierarchy. It contains data on the educational design of a course as well as on the students' learning progress within the course. Two main sources of this level data are a course syllabus/guide and LMS where the corresponding e-course is implemented. We now describe each block more precisely.

We start with describing *educational design data*. The structure of a certain course included in the curriculum of an educational program is built on the basis of reverse design technology, that is, on the basis of the declared learning outcomes. The course instructor determines a set of necessary and sufficient procedures for developing the declared learning outcomes, then selects the learning format, teaching strategies, methods, and techniques. The learning format admits various levels of involvement of the electronic environment into the learning process: traditional in-class learning with e-resources support, blended learning, or pure e-learning. When using an electronic environment as a web support for learning, the content of the e-course is limited to pieces of theoretical material aimed at refreshing the key points of the delivered lectures and assignments which can indicate whether the declared learning outcomes have been attained. The blended learning format requires implementing, in the e-course, all the necessary materials for students' pre-class and post-class independent work, while the pure e-learning format requires a comprehensive set of materials sufficient for self-administered learning. The set of teaching methods and techniques used by the course instructor has an influence on how well the knowledge and praxeological components of the declared learning outcomes are being developed.

Regarding *LMS digital footprint data*, we first note that most of the subjects taught in HEIs at present are provided with e-courses. Mid-tier HEIs mainly use e-courses developed and implemented in their existing information infrastructure, based on various LMS (Moodle, Blackboard, Canvas, eFront, etc.) with rare exceptions of already developed courses available on public online educational platforms (Coursera, EdX, Skillshare, Udemy, etc.). In the first case, the course developer is free to define the structure of the course, develop test materials, and determine the monitoring procedures and timeline. When using external courses, the course instructor needs to find an adequate e-course for their subject to make sure that it can develop the declared learning outcomes. Any LMS provides an administrator (a course instructor) with tools such as access to the course grade log and event log which contain data on student activity in the e-course. In Moodle LMS, the gradebook tool provides information about students' current scores for each assessed element as well as the final grade for the course. At the same time, the scores usually come without grading dates. This results in a situation where, in the case of changing grades (e.g., when a student has had several trials to complete the assignment), the scores of earlier trials become overwritten with that of the last one. However, for the purpose of monitoring the learning dynamics of a student, it is necessary to keep records (scores and dates) of all the trials. On the other hand, LMS can store more detailed data depending on the component of the e-course. For example, for a certain test element, LMS additionally stores the start and finish times of all test attempts, the times of entering each answer, the scores for each

question of the test, etc. The event log captures all user transitions within the e-course, and thus keeps a record of all student actions. Each event log entry contains identification data for the user performing a certain action (*User ID*), the name of a particular course item the user was interacting with (*Event Context*) at a certain moment, the type of the item the user was accessing (*Component*), and information about the action being taken (*Event Name*). For a given LMS, there is a fixed number of actions the user can perform with each type of the e-course element. Using the event log data, it becomes possible to derive various universal indicators from a student's digital footprint in an electronic environment, e.g., the number of views of a certain element in a given e-course, student performance on a particular kind of assignment, and the total number of scored points scored. It is also possible to obtain various derivatives of these indicators, such as the dynamics of a certain indicator over a certain period, the duration of periods of inactivity in the e-course, etc.

### 3.2. Education Database Framework

The key objective of the education database design is that it should simplify the processes of developing LA solutions and implementing a data-driven approach to managing the educational process. Figure 2 shows the architecture of the database. The entire data flow can be decomposed into the following levels: the level of data collection, the level of data storage, the level of data analysis, and the level of data usage. Each level contains various procedures for data processing and data transmission. We proceed with a more precise description of each level.

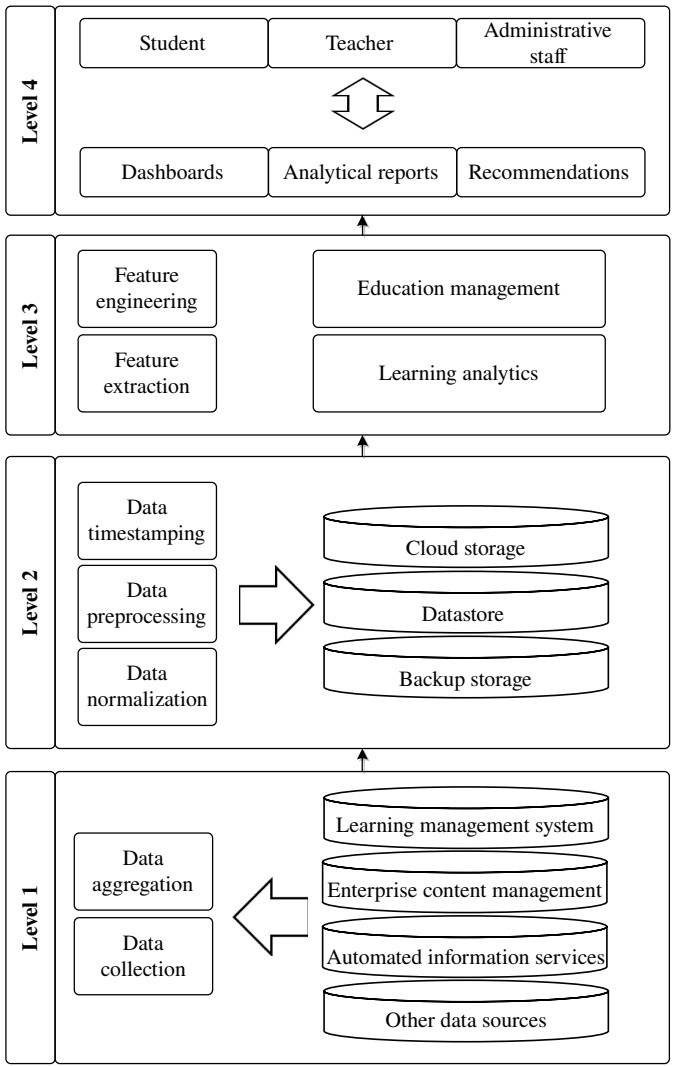

**Figure 2.** Architecture of the education database.

### 3.2.1. Level 1—Extracting Data

At the data collection level, education-related data are collected and accumulated from various sources, such as the LMS, management information systems, educational environment, and various digital services of a HEI. In SibFU, the main data sources are the services [36–39,41,42,44] described in Section 2.1. At this level, the following data processing and transmission processes are implemented: extracting and aggregating data from various sources into a single database, identifying and correcting data errors and inconsistencies (data cleansing), bringing data to a single format and structure (data standardization), and transferring data to a storage for further processing and analysis.

### 3.2.2. Level 2—Storing Data

After data extraction, it is necessary to store and organize data at the physical layer. This is the most important level in terms of defining the design of the education database. At this level, the database is being built and configured by means of creating tables and establishing relationships between them, distributing access rights to ensure data protection, and normalizing data by eliminating redundancies and inconsistent dependencies to increase database flexibility. Upon completing the configuration procedures, data can be loaded and stored. It is then possible to manage access rights, perform a backup, and monitor and optimize database performance. At this stage, it is necessary to comply with the principles of student centeredness, data continuity, and data consistency. The use of enterprise cloud storage is another important point that can simplify the process of data exchange and synchronization between HEI information systems and services. Additionally, it provides the configuration of data storage settings for solving certain LA or education management tasks.

### 3.2.3. Level 3—Analyzing Data

To make the extracting of valuable information possible, stored data should undergo various transformations. Solving the problems of LA or managing the educational process requires a thorough selection of the features most appropriate for solving a specific kind of problem, such as regression, classification, clustering, visualization, etc. This research-intensive stage requires serious analytical work involving statistical, machine learning, data mining, or other methods in order to gain valuable knowledge and understanding about educational activity, student performance, their learning needs, etc. At this level, university research teams use either ready-made solutions (e.g., Microsoft Azure ML, Google Cloud ML, RapidMiner, SAS) or custom developed software (data analytics applications are typically developed in Python/R).

### 3.2.4. Level 4—Using Data for Decision-Making

The top level is the human–computer interaction level. The data are presented to a certain type of stakeholder of the educational process (student, teacher, administrative staff member, top manager) in a readily comprehensible form and with a user-friendly interface. Usually, data are represented in the form of graphs, charts, tables, dashboards, reports or other forms that visualize results of analytics, allowing target recipients to quickly understand the contained information and make informed decisions. For instance, administrators can address issues related to streamlining the process of learning, course instructors can adjust course materials based on student performance results, and students can find recommendations for mastering their degree programs.

### 3.3. Cybersecurity and Ethical Frameworks

Taking into account the continuously increasing amount of stored data and growing cybersecurity threats, it is necessary to apply appropriate measures for data protection. Thus, the problem of ensuring cybersecurity at all levels is of primary importance in the process of designing and building an education database.

The technical aspects of cybersecurity include the prevention of unauthorized access to data and protection against external threats, such as malware attacks, to ensure the confidentiality, integrity, and availability of sensitive information. For this purpose, it is necessary to apply mandatory protection procedures: differentiating access rights for different groups of users, using safe up-to-date authentication and authorization methods, using data encryption protocols, making regular backups and having a data recovery plan in case of software/hardware failure or loss of information, ensuring physical security for server systems, and updating software on a regular basis to prevent the exploitation of vulnerabilities. All these procedures are provided by SibFU technical assistance to comply with national cybersecurity standards and regulations. In addition, it is necessary to take into account possible issues that may arise when integrating the database with various information systems and services. The process of integration must be accompanied with an investigation into the possible software/hardware interaction vulnerabilities and a description of the protective measures and procedures.

Along with technical aspects of data protection, it is equally important to take into account data ethics. Working with personal data requires full compliance with data privacy regulations. The corresponding regulating documents are General Data Protection Regulation (GDPR) [46] for the European Union, and the Federal Law No. 152-FZ on Personal Data (Personal Data Law) [47] in Russia. Both documents regulate the processing of personal data, establish security requirements, and grant certain rights to data subjects. The implementation of GDPR and the Federal Law No. 152-FZ in the educational context requires that the processing of personal data adheres to the principles of lawfulness, fairness, transparency, purpose limitation, data minimization, accuracy, storage limitation, integrity, and confidentiality with the responsibility lying with the controller (the body that determines the purposes and means of the processing of personal data) to comply with the mentioned principles. Educational institutions in Russia must comply with the requirements of the Federal Law No. 152-FZ, as well as take into account the principles of GDPR, especially when interacting with data subjects from the European Union or transferring data abroad. Therefore, it is necessary to take into account the requirements of both regulations to ensure the confidentiality and security of personal data when developing an education database.

Note that the mentioned regulations also apply to technical staff who have access to personal data due to their responsibilities. For instance, when using pieces of data for software testing purposes, one should use depersonalized data whenever it is possible.

Finally, all the aspects mentioned in this section should be reflected in the user requirements for the database users. In particular, the policies for information usage, requirements for user passwords complexity, frequency of passwords updating, and user authentication types should be established.

Compliance with the mentioned regulations and requirements is crucial for ensuring a higher level of data protection, which is a key component of the effective management of an education database and information systems interacting with it.

### 3.4. Mock-Up of the Education Database

By means of analyzing the samples of student education-related data obtained from SibFU information systems described in Section 2.1, we were able to reveal the typical issues of these data, including errors, incompleteness, inconsistencies, and inaccuracies due to human factors. Following the principles described in Section 2.2 based on the notion of educational history, we determined the structure of the database that can ensure the considerable reduction of these issues and provide convenience for developing LA and educational management solutions. The database was designed based on the principles of a relational database design using Microsoft SQL Server 2022, Microsoft Server Management Studio 19, and DB Designer [48] for defining the database structure. The database schema is given in Figure 3.

We now describe the key features of the database design.

According to the student centeredness principle, all the tables of the database are connected via the key field, *student_id* (the unique student identification code). This is also reflected by placing the *Student_personal_data* table in the center of the schema.

The data continuity principle means that the dates of various types of events (e.g., *exam_date*, *exam_retake_date*, *order_date*, *entrance_date*) are stored in all the student-related tables of the database.

Taking into account the fact that different information systems may contain data that duplicate each other, to comply with the data consistency principle, we performed a normalization step by removing data redundancies. The key fields of the database connecting different tables are: student ID (*student_id*), educational program ID (*edu_program_id*), curriculum ID (*curriculum_id*), course ID (*subject_id*), and e-course ID (*e-course_id*).

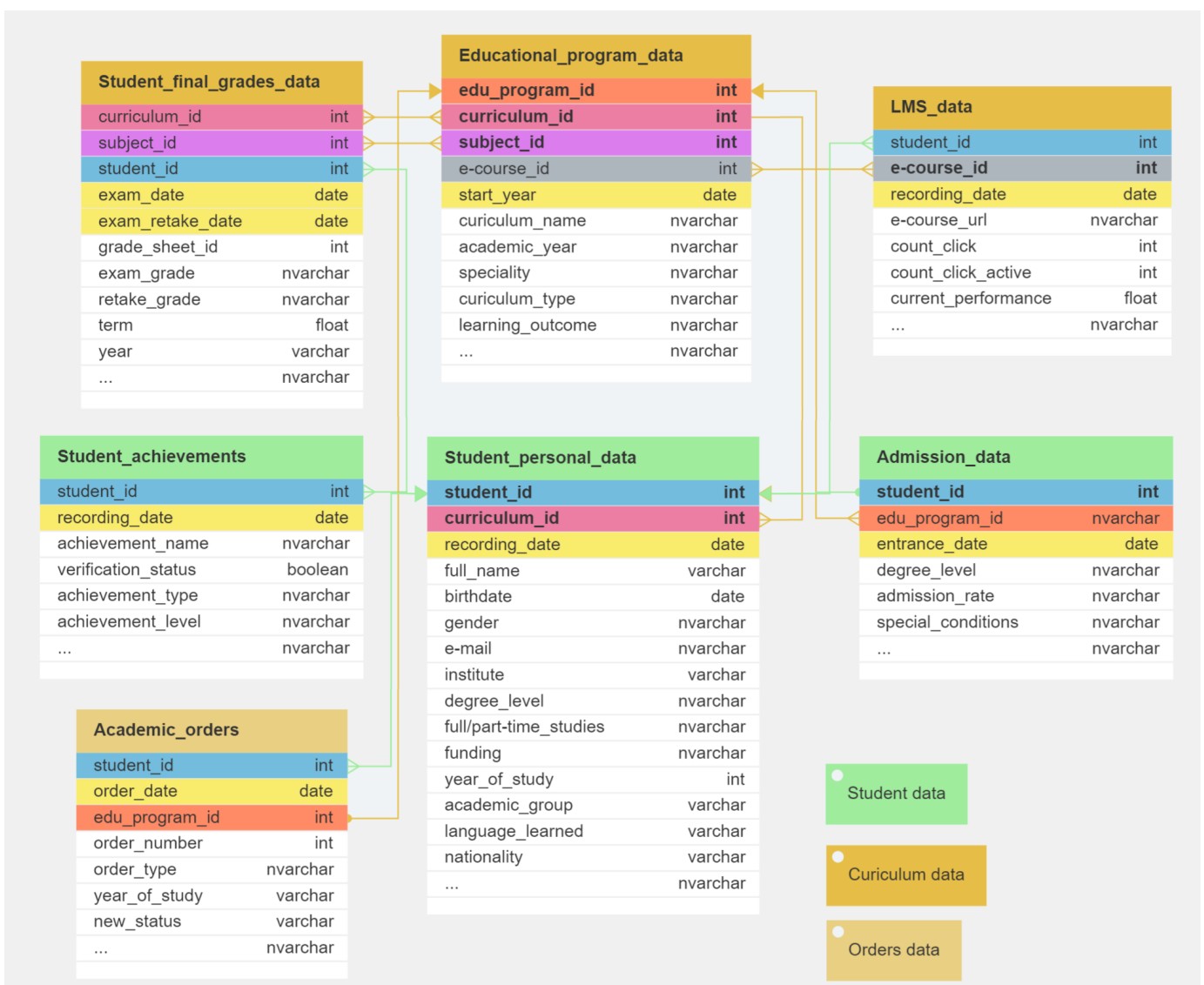

**Figure 3.** DB EduHistory database schema.

We illustrate the idea of normalization with the following example. When a certain individual first appears in the university admission information system as an enrollee, a unique student identification number (*student_id*) is issued. This ID never changes, and it is associated with the individual's personal data in the corresponding *Student_personal_data* table. This table contains the student's name, birthdate, etc. At the same time, the other university information systems may contain student personal data also, which can bring

various issues. First, information systems store redundant data that duplicate each other. Second, there might be errors (typos) due to human factors in some database tables which may yield conflicts between different information systems. Third, personal data (such as last name) may be changed, and the updates in different information systems may be performed at different times which, again, may lead to conflicts. The normalized university database keeps the personal data of a student in a single table, thereby avoiding data duplications. The student ID field links most of the database tables and university information systems.

Upon the enrollment of a student in a certain educational program, a unique educational program code (*edu_program_id*), one of the key fields in the database, is associated with a student ID. The other data such as degree level, full/part-time studies, and special conditions (based on medical record), if they exist, are stored in the *Admission_data* table.

Based on the integration of the *Plans* and *1C: Document Management* information systems, data are imported into the *Student_personal_data* table of the database where student personal data are stored: student ID, name, birthdate, gender, etc. It is connected with the *Educational_program_data* table via the educational program curriculum ID (*curiculum_id*) field. This table contains data related to the educational program and the corresponding curriculum, defining the current training track of a student.

The data on student achievements (*Student_achievements* table) obtained from *My SibFU* service are linked to the *Student_personal_data* table by the *student_id* field. The achievements table stores data on achievement name, type, level, confirmation status, and the time the record was added (*recording_date*).

The data on student learning activity in *e-Courses* LMS are imported into the *LMS_data* table. These data include the number of points scored (with timestamps identifying when they were scored), number of trials to complete a certain task, number of active clicks, total number of clicks, etc. Each course has its unique ID (*e-course_id*) linking the *LMS_data* table with the *Educational_program_data* table. *LMS_data* table is also linked with the *Student_personal_data* table via the *student_id* field.

Thus, the integration of data from various university information systems into a coherent whole allows one to design a database which can essentially simplify the solution of a wide range of LA and data-driven management tasks.

We now consider examples of reports that can be obtained from the DB EduHistory database using certain SQL queries with automatic processing of the results.

**Example 1.** *Table 1 shows the grade distribution for the final exam of the Calculus-II class for a certain academic group of first-year bachelor students (N/A means "not attended").*

**Table 1.** Grade distribution for a certain Calculus-II class.

| Final Grade | Number of Students | Percentage |
|:---:|:---:|:---:|
| A | 6 | 24% |
| B | 8 | 32% |
| C | 5 | 20% |
| F | 2 | 8% |
| N/A | 4 | 16% |
| TOTAL | 25 | 100% |

**Example 2.** *Table 2 shows several records of the report on the academic history of a certain student typically used by administrative personnel.*

**Table 2.** Academic history of a student.

| Student ID: | 12345 |
|---|---|
| Enrollment date: | 11 August 2020 |
| Last name: | Lastname1 |
| First name: | Anna |
| Academic status: | Enrolled |
| Degree level: | Bachelor |
| Major: | Computer Science |
| Academic group ID: | CS 20-11 |
| Academic status update 1: | Academic leave |
| Academic status update 1 (date): | 23 April 2021 |
| Academic status update 1 (reason): | medical |
| Academic status update 2: | Enrolled |
| Academic status update 2 (date): | 5 February 2022 |
| Academic status update 2 (reason): | Return from academic leave |
| Major update 1: | Software Engineering |
| Major update 1 (date): | 5 February 2022 |
| Major update 1 (reason): | Student request (approved) |
| Academic group ID update 1: | SE 21-05 |
| Academic group ID update 1 (date): | 5 February 2022 |
| Academic group ID update 1 (reason): | Change of major |

*3.5. Database Evaluation Plan*

Assessing the quality of a database after it has been developed is an important step to verify that it meets the goals of its development, security requirements, and user expectations. This phase includes an evaluation of its performance parameters, efficacy, usability, acceptance, and satisfaction. To evaluate these components, we propose using the following plan.

- *Receiving feedback from technical support specialists.*
  Technical experts verify the validity and integrity of data, conduct workload benchmarking under various usage scenarios, and identify and fix hardware and software problems. Interviewing experts can help in identifying issues that end users of the database might experience in assessing the security of the system. Additionally, periodic checks of the database allow one to detect errors, data redundancies, and other problems that may affect database performance.
- *Receiving feedback from end users.*
  Interviewing students, academic staff members, and administrative personnel allows one to understand how the system meets their needs and expectations. This may help to evaluate the usability of the database, its acceptance among users, and their satisfaction with its usage. The timely resolution of identified problems and the implementation of improvements can significantly improve end user experience.
- *Performance and efficacy testing.*
  Measuring the time spent executing various queries allows one to evaluate the database performance and identify its possible bottlenecks. Another indicator that one may consider is the number of queries (complexity of a query) needed to obtain the desired information. When implementing a new database, it is important to ensure that the information necessary for managing the educational process is easily accessible. Assessing this parameter allows one to determine the usability and efficacy of the database.
- *Solvability of LA problems.*
  One of the goals of introducing a new database is to widen the possibilities of using LA solutions at the university, so it is important to evaluate whether the new database makes it possible to solve LA problems that were impossible or hard to solve with the previous database structure. Moreover, one can perform a comparative analysis

of the quality of forecasting systems based on the data obtained from the database. This allows one to determine the advantages and disadvantages of the new database compared to similar solutions (ready-made or previously used at the university). For example, using the proposed database, that provides easy access to new features and their derivatives, can significantly improve the learning success prediction quality for a forecasting system that was built based on the former database(s).

### 3.6. Examples of Applications

3.6.1. Educational Programs Quality Assessment

Obviously, student performance on a certain educational program or a course can depend not only on their abilities or attitude towards their studies, but also on the educational design of a program/course, teaching methods, and proficiency and personal characteristics of course instructors. We consider the problem of determining the quality of the educational program in terms of how well its content, structure, and educational design contribute to the successful achievement of the declared learning outcomes by students mastering the program.

As a first step of researching this said problem, we consider the curriculum of the educational program as a main document that defines its structure, content, and educational design. At this point, we neither distinguish between similar courses (of a similar number of credits) of different educational programs implemented in different HEIs nor take into account the impact of course instructors on the quality of the course. We thus focus on studying the curriculum, its content (courses, internships, qualification thesis), and its structure (their consistency, order, interdisciplinary links, etc.) for the purpose of identifying the relationship between the design of the educational program and its quality.

In reference [49], the authors propose a model for representing the curriculum as a simple weighted undirected graph with vertices representing academic disciplines (implemented as courses in a certain educational program) and edges representing interdisciplinary links. The interdisciplinary links are defined based on their labor intensity (measured in number of credits assigned to the corresponding course) and learning outcomes formed by the corresponding courses. The authors propose possible integral characteristics of graph representations, on the basis of which it is possible to estimate the quality of the corresponding educational program, as well as perform a comparative analysis of different educational programs. The visual representation of interdisciplinary connections helps us to better understand the structure of the curriculum, identify disciplines with a maximum number of interdisciplinary connections, and decompose the curriculum into clusters of the mostly interconnected disciplines.

Although it is difficult to define precisely what the quality of an educational program is, it is still possible to determine whether a program is finely designed or has serious issues. For this purpose, one can consider such characteristics as the number of applicants per place (state quota), average entrance test scores, information on incoming and outgoing student migration (with regard to a certain educational program), the distribution of final course grades, and the numbers of exam retakes. Furthermore, it is possible to survey students on their satisfaction with various components of the educational program they master: schedule, course content, proficiency and attitude of the course instructors, and opportunities provided by the program (internships, career perspectives, etc.). Another important parameter is the share of graduates of the program who successfully found a job connected to the attained profession within a limited period of time after their graduation. It also makes sense to consider such data on the inter-university level by comparing similar educational programs implemented in different HEIs.

After maintaining the quality assessment of an educational program, one can pose the problem of identifying features of its curriculum connected to its advantages or weaknesses. This approach may lead to the development of tools for designing an "optimal" curriculum for a given educational program. Here, optimality means the students achieving the learning outcomes of an educational program in a better way. At the present moment, the

problem of constructing a curriculum of an educational program by means of LA tools is at its very beginning [50,51]. At the same time, results in this field may lead to breakthroughs in data-driven learning management.

### 3.6.2. Prediction of Student Dropout

Based on existing studies on predicting student dropout, we make a hypothesis that adding features reflecting the dynamics of student performance over time contributes to a significant improvement in the quality of the models used. To test the hypothesis, we carried out the experiments as follows.

Data on 46,535 students were extracted from SibFU information system databases (we used data for five consecutive years). The dataset was randomly split into training and test datasets with the ratio of 70:30. Initially, we built predictive models on common student characteristics which are usually gathered in HEIs and used for predicting performance and dropouts:

1. Socio-demographics: age, citizenship, preferences for entering HEI, funding source(s);
2. General learning process characteristics: form of study, faculty, year and semester of study, degree level, specialty, individual study plan;
3. Academic performance characteristics for three previous semesters: average final grades, number of exams.

Since there were complex relationships between variables in the obtained dataset, we chose the following powerful ensemble of algorithms for prediction—Random Forest Classifier and Catboost Classifier, which is particularly effective for handling datasets with categorical features. Our dataset was exactly the same case as the information about curricula and learning conditions was provided mainly by categorical variables.

The dropout rate of students was 4.9%. As the data were highly imbalanced, we fitted the models on the training dataset using oversampling. All models were trained with default hyperparameters (sklearn 1.2.1, catboost 1.0.6).

However, the performance of the models on the test dataset (Table 3) was unsatisfactory from the viewpoint of at-risk student detection (with recall equal to 0.615 and 0.545, respectively), although accuracy was high due to the class imbalance.

**Table 3.** Classification performance metrics on the test dataset.

| Metrics | Catboost Classifier | | Random Forest Classifier | |
|---|---|---|---|---|
| | Without Education History | Including Education History | Without Education History | Including Education History |
| recall | 0.615 | 0.958 | 0.545 | 0.930 |
| precision | 0.737 | 0.913 | 0.794 | 0.887 |
| F1-score | 0.671 | 0.935 | 0.646 | 0.908 |
| ROC-AUC | 0.967 | 0.999 | 0.968 | 0.999 |

To extend the data with detailed information about student learning history, we extracted historical data from the education databases' backups and applied them to the bases of orders for the university. We added the following characteristics to the initial list of predictors:

1. Relocation history: number of academic leaves, transfers and dropouts and their reasons;
2. History of grades: average grades before and after retakes for three semesters, numbers of retakes.

One can see the increase in the performance metrics (Table 3) and the improvement in the ROC curves (Figure 4). The most impressive growth was shown by recall (which increased up to 0.958 and 0.930, respectively).

The added predictors are of high predictive power. In Figure 5, one can see the predictors with the highest feature importance (both for Catboost and Random Forest). The

dark green bars of the histogram correspond to the variables included both in the initial and the extended models, whereas light green bars—only in the extended versions.

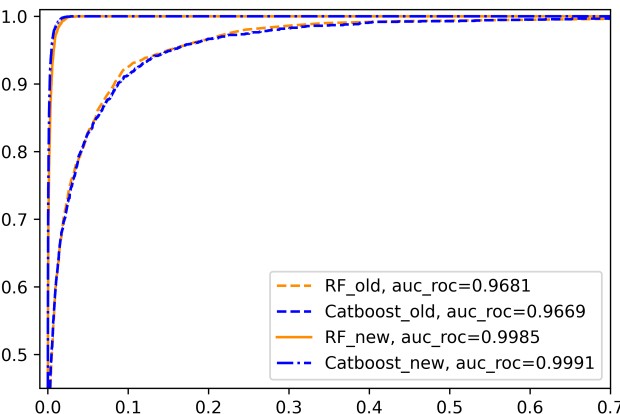

**Figure 4.** ROC curves for the Random Forest and the Catboost classifier trained on data without education history (RF_old and Catboost_old) and on data including education history (RF_new and Catboost_new).

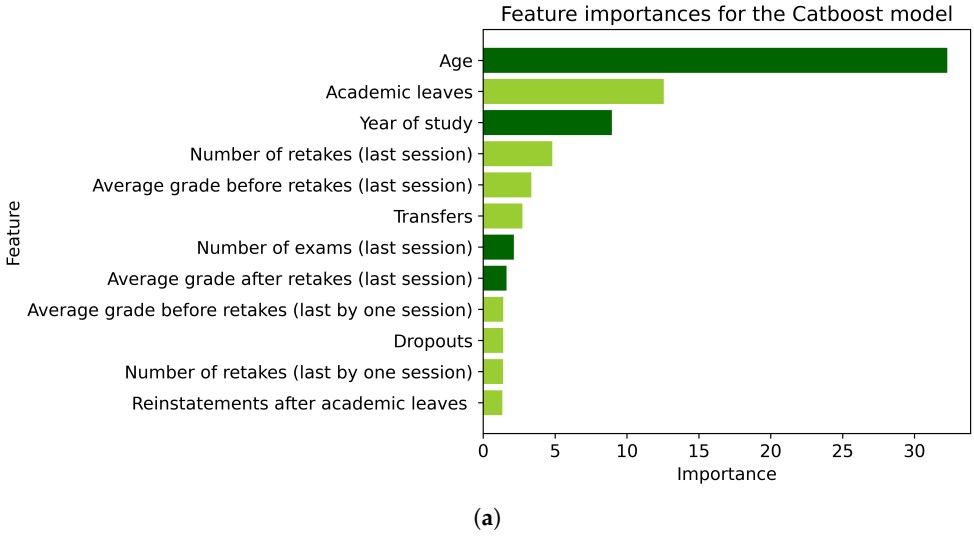

(**a**)

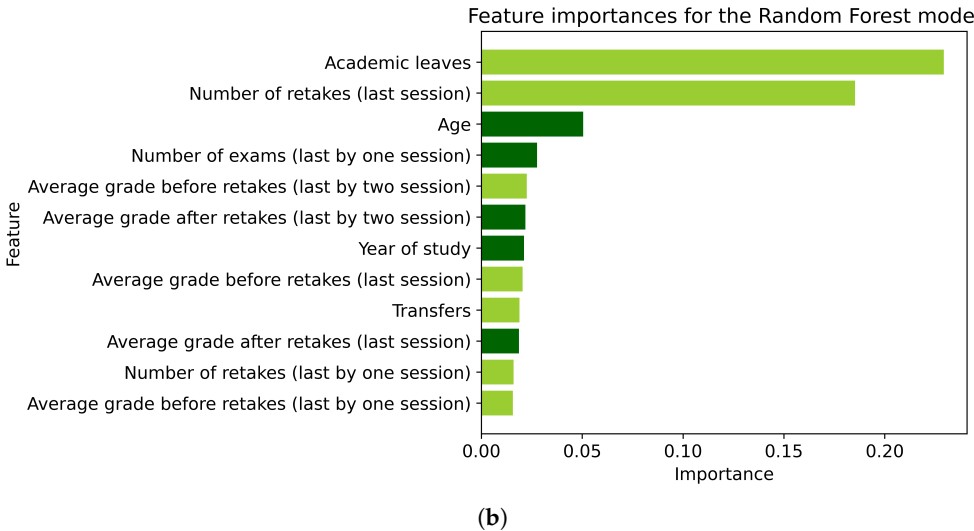

(**b**)

**Figure 5.** Features with the highest importance for: (**a**) the Catboost model (**b**); the Random Forest model.

New predictors incorporate information about dropout risk factors (as all relocations and retakes indicate difficulties with current studies), thus the extended models are better at detecting at-risk students.

## 4. Discussion

The development and implementation of the proposed educational database is a part of the development of the digital infrastructure of a HEI. The latter is a process that consists of implementing ready-made IT solutions, upgrading existing services, and developing new ones for custom purposes. In a university, this process usually takes an extended period of time and can meet resistance at various levels including faculty members, heads of administrative units, etc. Moreover, new university leaders and executive teams inherit existing infrastructure and take further steps on its development with their own vision that often differs from that of their predecessors. This results in various issues regarding using data for the development of data-driven decision-making solutions. These issues, straightforward for learning analysts, may remain unnoticed or unperceived by top managers who are decision-makers in most of the cases.

For this reason, the implementation of new concepts should not just start with the approval of chief information officers or chief technology officers—they or their representatives should be members of the transformation team during the entire implementation process, together with lead developers, learning analysts, and representatives of the end users.

As an example, we refer to our work on the predictive models described in Section 3.6.2. The models were built for the SibFU Early Alert System that is currently under development. Working as members of the Early Alert System developer team, we encountered some challenges related to organizing the work. In particular, we experienced the problem of university staff resistance to change which is quite typical (see, e.g., reference [17]).

Furthermore, there are problems specifically related to the implementation of the previously mentioned principles of the education database design. In the case of SibFU, the transition from the existing database architecture, which does not allow for the storing of temporal indicators for all student data, to an architecture that does, requires a significant reworking of nearly all university data collection and storage systems. Therefore, the problem of building an adequate digital infrastructure transformation team is of primary importance.

Another important aspect of the problem of the secure and efficient use of education data by a HEI is the technical design of the education data ecosystem. The majority of HEIs initially started with disparate information systems designed for specific tasks. The integration of such systems is rather problematic from a technical point of view. It often leads to inefficient resource allocation, information security concerns, and work duplications.

Many HEIs practice replacing their existing information systems with enterprise resource planning (ERP) systems that integrate business processes, functions, and data [52]. Among the most well-known ERP systems are SAP, Oracle, and PeopleSoft, which offer special software products for universities. The most widespread enterprise software for HEIs in Russia is *1C: University* [53]. Along with the obvious benefits of ERP systems, there are also barriers and limitations. These systems are rather software- and hardware-demanding, and they are quite expensive. Moreover, adjusting them to the specifics of particular tasks requires functional rework, which can cause functionality problems during further maintenance or during the installation of newer versions or automatic updates [54].

In our opinion, it is technically easier to implement the proposed conceptual approach for building an education database within a HEI's own digital ecosystem rather than using ready-made ERP solutions designed for HEIs. But, undeniably, in both cases, creating an actionable education database, as well as the development of LA tools based on it, requires the involvement of various university resources and stakeholders, including administrative and teaching staff, software developers, and data analysts.

As for the internal use of data, in addition to data privacy and data security issues, there arises a question of the ethical use of decisions based on education data [55–57].

Student educational history is built on the raw data of digital footprints which require serious preprocessing before analysis and interpretation (particularly the data on user actions on e-learning platforms). The algorithms and approaches used in this preprocessing to construct features for machine learning models are a significant source of possible distortions and loss of information [58]. Thus, there is a risk of making incorrect decisions based on information that is not an indisputable fact but rather a product of multiple automatic processing and often not entirely objective interpretations.

To solve this problem, it is important, first of all, to ensure the complete transparency of the data collection procedures and all decision-making stages related to educational activities. Next, the models used in data analysis should be interpretable in order to identify causal relationships between the characteristics of the learner's personal profile, the parameters of the learning environment, and the learning process. Finally, the decisions based on educational history should be of a recommendatory nature, not prescriptive, so that they do not limit students' possibilities.

To give an idea of the possible directions of further research, we note that the digital personality portrait of a student is connected closely enough to their digital educational history. Clearly, personal characteristics have a direct impact on the processes of cognition, learning, and, as a result, on the way a student interacts with an e-learning environment. This works in the other direction as well: digital educational history comprises data from which it is possible to extract information on personal learning and cognitive styles which are derivatives of personal characteristics. This makes it possible to consider the problem of reconstructing personal characteristics by means of digital educational history data.

At the present moment, the prototype of the educational database presented in this work is ready for pilot implementation. The transformation project team is currently being built. The inclusion of a wide range of stakeholders (analysts, developers, chief information officer, representatives of the student community, top management, academic and administrative staff, etc.) is aimed at achieving flexibility in the resulting solution, which is difficult to achieve using available ready-made solutions.

The pilot implementation is planned to be carried out based on the principles of business process implementation and the assessment plan provided in Section 3.5. The key stages of the pilot implementation will be functional and performance testing, security assessment, and comparative analysis with the former database. The main goals of the functional testing are checking the correctness of data processing and the compliance with the principles given in this work. Performance testing is aimed at determining how efficiently the database can access data and process various queries. Security assessment determines how secure the database is against unauthorized access and whether cybersecurity and ethical regulations are met. Finally, a comparative analysis with the previous database involves a comprehensive assessment of the positive effect of implementation based on qualitative and quantitative metrics.

These procedures will help in determining how efficient the proposed prototype of the educational database is, and in estimating its impact on solving LA and educational data-driven management problems.

## 5. Conclusions

In the context of the steady trend of the massification of higher education, modern universities face the challenging task of the multi-objective optimization of the educational process, which includes enhancing student experience, updating educational programs, engaging teaching staff, improving document workflow, etc. The proposed education database may serve as a background for solving this kind of optimization problem, both from the HEI and student points of view.

The paper addresses the issue of designing an education database that is responsive to all kinds of educational management and LA tasks. The application of the proposed

conceptual approach for developing the architecture of this database can enhance the data-driven management of the educational process, with a particular focus on learners.

The proposed designing principles of student centeredness, data continuity, and data consistency are promising for elevating LA in universities to a new level: from simple monitoring to advanced analytics and digital learning support systems. Such systems can make the process of learning more transparent, which can be beneficial for the various stakeholders of the educational process, including students, teaching staff, and top managers. The implementation of these principles may require significant changes in the existing database infrastructures of HEIs, as well as training the university staff who will use the information systems related to the database.

**Author Contributions:** Conceptualization, T.A.K., R.V.E., A.A.K. and T.V.Z.; methodology, T.A.K. and A.A.K.; software, R.V.E. and T.V.Z.; formal analysis, T.A.K., R.V.E. and T.V.Z.; writing (original draft preparation), A.A.K., T.A.K., R.V.E. and T.V.Z.; writing (review and editing), A.A.K. and T.A.K.; visualization, T.V.Z.; supervision, A.A.K.; project administration, A.A.K.; funding acquisition, A.A.K. All authors have read and agreed to the published version of the manuscript.

**Funding:** This research was funded by the Russian Science Foundation, grant number 22-28-00413.

**Institutional Review Board Statement:** The study was conducted in accordance with the Declaration of Helsinki, and approved by the Institutional Review Board of the School of Space and Information Technology of Siberian Federal University (10/2, 28 May 2021).

**Informed Consent Statement:** Informed consent was obtained from all subjects involved in the study.

**Data Availability Statement:** The samples of depersonalized data used in this study can be provided upon request at tkustitskaya@sfu-kras.ru.

**Conflicts of Interest:** The authors declare no conflict of interest.

## Abbreviations

The following abbreviations are used in this manuscript:

| | |
|---|---|
| DB | Database |
| ECM | Enterprise Content Management |
| ERP | Enterprise Resource Planning |
| GDPR | General Data Protection Regulation |
| HEI | Higher Education Institution |
| ID | Identification Code |
| LA | Learning Analytics |
| LMS | Learning Management System |
| SibFU | Siberian Federal University |

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
