# Peer review of "Designing an Education Database in a Higher Education Institution for the Data-Driven Management of the Educational Process"

_education, doi:10.3390/educsci13090947_

Round 1

Reviewer 1 Report

Overview

The article has an important topic for scientific research both educational and technical, i.e., to build technical infrastructures (Databases) to support the Universities and their educational role in society and economy for the next generations. In this regard, the data collected manually (bureaucracy) or automatically (digital devices) related to the students (e.g., sociodemographics, educational outcomes, notes from teachers, data from a smartwatch, etc.) can improve the educational approaches and its impact on the student’s life and outcomes. Thereby, the authors aim to build a database that focuses on the complexity of the educational experience (not linear), adopting a comprehensive approach. This goal is now possible because of the new digital and technical development, for instance, artificial intelligence, data analytics, machine learning, multi-cloud architecture, etc. Despite the authors’ intentions, the article requires more work to be a scientific article.

Review

The article requires a more accurate division of contents. In the section of “results”, there are references for literature (that should feed the discussion and/or the introduction) and the methodological options (namely, the principles). Thereby, the section related to the “methodology” doesn’t explain the “materials” (e.g., databases analysed and their contents) and methods (e.g., software to build a new database, methods to analyse the existing sources, the review of other sources, as explained in the lines 99 and 100). Additionally, the section on “results” doesn’t include critical topics related to designing a database: architecture, cybersecurity structure (including data protection and ethical framework), data analytics methods, user requirements, and, especially, the theoretical framework under the architecture. In the discussion, it is missing the discussion about if the “new” database works or not, and what is the impact (or expected impact); in this regard, it’s necessary an evaluation plan/tool to validate the database (efficacy, feasibility, usability, acceptance, satisfaction), for example, a pilot study.

My suggestion for topics to develop in each section is:

Introduction: The HEI education challenge in a digital society; The opportunities for the HEI related to digital and technical innovation; The role of the Databases for education and students’ management in HEI.

Material and Methods: Description of Existing Databases and their owner, the SbiFU (only the data used); Description of the Principles of Designing; Description of the Software adopted; Description of the Pathway or steps done to develop a new database.

Results (this is critical): Theoretical Framework (why each data is related to another); Architecture (Graphic); Cybersecurity framework (technical); GRDP and Ethical framework; User Requirements; Mock-up or a Prototype (Graphics, textual descriptions, pictures); Evaluation Plan (it could be a plan).

Discussion: Lessons learned, Limits, Potentials, Next Steps.

I would suggest a review by a professional.

Author Response

Dear colleague,

On behalf of our team, I would like to thank you for your valuable comments as they prompted us to perform a significant revision of our paper which, I hope, made it more meaningful, comprehensible and reader-friendly.

Since your comments are rather a coherent whole then a separate points, I will try to describe the changes we have done, one by one, explaining how we addressed your points.

We significantly restructured our manuscript according to your suggestions. We have divided “Introduction” and “Materials and Methods” into subsections each of which addresses a certain point. The parts that are completely new (were absent in the initial version) are highlighted in violet. The text that was shifted from one section to another (e.g., Principles of design) is not highlighted, but the division into subsections can as well serve as a guide through our revisions.

I briefly go through the updated structure of the paper:
1. Introduction
1.1 explains the role of data in higher education, describes challenges and opportunities for HEIs
1.2 describes state of the art in learning analytics
1.3 describes typical problems related to data collection and usage in a HEI (partially new)

2. Materials and Methods
2.1 describes existing digital infrastructure (information systems) of the “host” institution, SibFU
2.2 explains principles of designing the database
2.3 describes the steps we have done in this research work (NEW)

3. Results
3.1 describes the levels of detail of education data
3.2 presents framework and architecture of the database (NEW)
3.3 presents cybersecurity and ethical frameworks, connects them with user requirements (NEW)
3.4 presents mock-up of the database (the quality of Figure 3 was improved), describes the data structure based on the principles from 2.2, gives examples (new) of the results of processing database queries.
3.5 describes the database evaluation plan (NEW)
3.6 gives examples of applications, 3.6.2 emphasizes improvements in machine learning models after implementing the proposed principles

4. Discussion
We discuss various issues that may arise on the way of transformation of digital infrastructure, including those based on our own experience. Give ideas on the future research and described our vision (partially approved by top managers, partially still under discussion) of pilot implementation of the database prototype.

Regarding language issues, I together with my colleague from linguistics department read through the paper and tried to reformulate the places that sound awkward. Please let me know if you fill something still can be improved

I hope this version addresses most of your suggestions. I’m sorry that we did not submit the revised version earlier, but there was a lot of work and we spent all this time trying our best to improve the manuscript.

Reviewer 2 Report

Article would benefit from additional example data about the student

and courses. 

For readers use, authors should add a few SQL queries relavant to the DB: how many students involved in a particular class, what is the grade distribution for the class, are some classes/instructors are harder than other.

Care should be taken into working with sensitive data like age, socio-economical status, please extend how this data is protected.

For drop-out application it is not clear if dataset has been split into train and test/validation dataset and if metrics are reported on test split. Please add details about train/test/validation dataset sizes.

Authors should report drop-out rate to help reader understand how severe is the class balance. Use of the accuracy metrics here is misleading, please remove it and use precision and recall and if possible add ROC curve.

Authors choose catboost and random forest models, but don't specify why these models were chosen or how hyper-paramenters were optimized, please extend the discussion of this section.

Author Response

Dear colleague,

On behalf of our team, I would like to thank you for your valuable comments that helped us to improve our paper.

First of all, I would like to note that our manuscript has undergone a significant revision based on the comments of Reviewer 1. It was considerably restructured. The parts that are completely new (were absent in the initial version) are highlighted in violet.

I will go now through your points explaining where they were addressed in the paper.

For readers use, authors should add a few SQL queries relavant to the DB: how many students involved in a particular class, what is the grade distribution for the class, are some classes/instructors are harder than other.

> Please, see the examples at the end of Section 3.4. We discussed your point and came to the conclusion, that for the reader it would be beneficial to see the results of SQL quieries rather then the quiery code.

Care should be taken into working with sensitive data like age, socio-economical status, please extend how this data is protected.

> Please, see Section 3.3 Cybersecurity and ethical frameworks. 

For drop-out application it is not clear if dataset has been split into train and test/validation dataset and if metrics are reported on test split. Please add details about train/test/validation dataset sizes.

> Please, see the added text (highlighted in violet) in Section 3.6.2, paragraph 2.

Authors should report drop-out rate to help reader understand how severe is the class balance. Use of the accuracy metrics here is misleading, please remove it and use precision and recall and if possible add ROC curve.

> Please, see the added text (highlighted in violet) in Section 3.6.2, paragraphs 5-6, Table 3, and Figure 4.

Authors choose catboost and random forest models, but don't specify why these models were chosen or how hyper-paramenters were optimized, please extend the discussion of this section.

> Please, see the added text (highlighted in violet) in Section 3.6.2, paragraph 4.

Reviewer 3 Report

1) In my opinion, more arguments should be given in the introduction for the creation of another database/system for universities, which enable "management of an effective educational process at a university". In addition, the research gap should be better documented in the introduction. Why is another new database needed and who would use it? What will this database bring? Does the implementation of this database mean that each student will have an individual learning path resulting from data analysis? What's new in the proposed database?

2) The article lacks an overview of international databases that are used by universities around the world and are used to manage an effective educational process at the university. The authors wrote: More and more universities are turning towards learning analytics and data-driven learning management (lines 629-631) - please elaborate in the article (please write more) - which universities implement new systems and databases? What problems do they face? What do they expect?

3) What are the practical implications of implementing a new/developed database? What benefits will the university achieve? What benefits will students get? How will the new database affect the quality of education? Why would a university use a new database? What does it mean that the new database takes education to a new level?

Author Response

Dear colleague,

On behalf of our team, I would like to thank you for your valuable comments that helped us to improve our paper.

First of all, I would like to note that our manuscript has undergone a significant revision based on the comments of Reviewer 1. It was considerably restructured. The parts that are completely new (were absent in the initial version) are highlighted in violet.

Before I go to specific comments, I would like to say that our motivation for writing this paper were difficulties with the data obtained from the university (SibFU) information systems when we worked on developing first an Early-Warning system for a specific course, and then, the university-wide Early Alert System. Based on the study of the samples of data, we came to the conclusion, that the data in the current structure is very hard to use for development of Learning Analytics solutions. It requires serios preprocessing with some peculiarities that could not be resolved automatically, and manual resolution in each case is something the university cannot afford. Based on this, we formulated the principles of database design following which, it will be possible to avoid the mentioned problems and develop the database with fine data structure that will enhance developing Learning Analytics solutions instead of being an obstacle to their development. Thus, in this work we propose rather the principles for designing a database, than "yet another database". We are pretty sure that these problems with data exist in a majority of universities possibly excluding the very top HEIs. We could find research articles on this very matter as what is usually meant by "educational database" is something completely different from what we propose. On the other hand, there are a lot of studies on information systems and Learning Analytics solutions (we cite the most relevant of them), but the problem is that in all this works very little is said about the data structure that lies behind those solutions. Nevetheless, we believe, that these questions are rather important, and this is one of the key ideas of our work.

As each your point consisted of a few questions, I will try to briefly comment on them and explain where they were addressed in the paper.

1) In my opinion, more arguments should be given in the introduction for the creation of another database/system for universities, which enable "management of an effective educational process at a university". In addition, the research gap should be better documented in the introduction. Why is another new database needed and who would use it? What will this database bring? Does the implementation of this database mean that each student will have an individual learning path resulting from data analysis? What's new in the proposed database?

> As I mentioned above, the research gap is that the data formats and structures typically used at HEIs do not meet the requirements of digital solutions that universities expect to have in the near future. This is discussed in Sections 1.3, 2.2, 3.4, Discussion. In 3.6.2 on a particular example we show how the proposed principles based on the notion of educational history can improve machine learning models. The concept of the database based on the proposed principles is a novelty.

2) The article lacks an overview of international databases that are used by universities around the world and are used to manage an effective educational process at the university. The authors wrote: More and more universities are turning towards learning analytics and data-driven learning management (lines 629-631) - please elaborate in the article (please write more) - which universities implement new systems and databases? What problems do they face? What do they expect?

> This is discussed in Sections 1.2, 1.3, and Discussion.

3) What are the practical implications of implementing a new/developed database? What benefits will the university achieve? What benefits will students get? How will the new database affect the quality of education? Why would a university use a new database? What does it mean that the new database takes education to a new level?

> The examples that show potential of the proposed approach are given in Section 3.6.1, 3.6.2. By a "new level" we mean technologies of personalized learning and data-driven management of an educational process which are still under research and development.

Round 2

Reviewer 1 Report

Dear authors,

Congratulations. I hope my contributions had been helpful for your review. The article could be published now. I would like to suggest a final reading to assess the quality of the English and text coherence.

Author Response

Dear colleague,

Thank you very much for your feedback! We have performed a final reading with numerous minor corrections aimed at improving language quality and text coherence.

Reviewer 3 Report

The authors responded to my comments and suggestions. The article has been supplemented. Thank you

Author Response

(The authors gave the same response as above.)
